# A Hierarchical Model to Predict Time of Flowering of Kiwifruit Using Weather Data and Budbreak Dynamics

**DOI:** 10.3390/plants13162231

**Published:** 2024-08-12

**Authors:** Jingjing Zhang, Maryam Alavi, Lindy Guo, Annette C. Richardson, Kris Kramer-Walter, Victoria French, Linley Jesson

**Affiliations:** The New Zealand Institute for Plant and Food Research Limited, Private Bag 92169, Auckland 1142, New Zealand; maryam.alavi@plantandfood.co.nz (M.A.); lindy.guo@plantandfood.co.nz (L.G.); annette.richardson@plantandfood.co.nz (A.C.R.); kris.kramer-walter@plantandfood.co.nz (K.K.-W.); victoria.eyre@plantandfood.co.nz (V.F.); linley.jesson@plantandfood.co.nz (L.J.)

**Keywords:** plant phenology, climate change, budbreak, flowering, predictive modeling, thermal requirements, kiwifruit cultivar, orchard management

## Abstract

Accurate prediction of flowering times is essential for efficient orchard management for kiwifruit, facilitating timely pest and disease control and pollination interventions. In this study, we developed a predictive model for flowering time using weather data and observations of budbreak dynamics for the ‘Hayward’ and ‘Zesy002’ kiwifruit. We used historic data of untreated plants collected from 32 previous studies conducted between 2007 and 2022 and analyzed budbreak and flowering timing alongside cumulative heat sum (growing degree days, GDDs), chilling unit (CU) accumulation, and other environmental variables using weather data from the weather stations nearest to the study orchards. We trained/parameterized the model with data from 2007 to 2019, and then evaluated the model’s efficacy using testing data from 2020 to 2022. Regression models identified a hierarchical structure with the accumulation of GDDs at the start of budbreak, one of the key predictors of flowering time. The findings suggest that integrating climatic data with phenological events such as budbreak can enhance the predictability of flowering in kiwifruit vines, offering a valuable tool for kiwifruit orchard management.

## 1. Introduction

The kiwifruit industry is the biggest horticultural sector in New Zealand, worth over USD 3 billion in annual gross sales [1]. Budbreak and flowering are the crucial stages of kiwifruit vine phenological development that mark the sequential progression from dormancy to active growth and reproduction [2,3]. Budbreak of kiwifruit vines is sensitive to climatic conditions, especially temperature or thermal times such as chilling unit accumulation and accumulated growing degree days [4], which affect dormancy and growth through cell metabolism, carbon accumulation and other biochemical processes [5]. Temperate fruit trees and crops require a period of cool temperatures (chilling unit; CU accumulation) followed by warmer temperatures (growing degree days; GDDs) in spring to initiate budbreak and flowering [6]. The knowledge of the annual timing of phenophases and their variability can help decision-making in orchard management, such as timing of chemical applications or the introduction of honey bees as the key vector for pollination [7,8]. Therefore, prediction models for kiwifruit flowering time and the variation in flowering are important support tools for the implementation of cropping schedules and management operations, especially with the risk of the shifts in local and regional atmospheric patterns with potential climate change in the future.

Early research on *Actinidia chinensis* var. deliciosa ‘Hayward’ in New Zealand has established a baseline understanding of the correlation between temperature and the phenological stages of kiwifruit, revealing a direct influence on the timing of budbreak and flowering [9,10,11,12]. For example, standard pistillate cultivars such as ‘Hayward’ require a particularly long frost-free growing season of ~220 days and 750 to 800 CUs for satisfactory production [13]. An increase in flower numbers was found when ‘Hayward’ vines were exposed to temperatures of 7 °C (45° F) or lower [14,15]. In New Zealand, hydrogen cyanamide (HC) is often applied to kiwifruit vines to increase budbreak and flowering and to synchronize flowering within an orchard [16]. Therefore, the orchard managers need a reliable prediction of the budbreak and flowering times to decide on the time window when treatments like HC should be applied. Application of such budbreak enhancers will influence the timing of budbreak and flowering.

The biological and genomic mechanisms of budbreak, dormancy and flowering in kiwifruit have been recently reviewed [3]. While understanding the biological factors and mechanisms underlying flowering is important in developing new budbreak enhancers, management techniques and new cultivars, understanding regional effects, responses to weather, and climate trends is also required to develop decision-making support tools for orchard management. Thus, there is a need for phenomenological models that focus on the timing prediction, uncertainty, and weather or climate impacts at the orchard, regional or national scale, with practical data collection strategies such as weather stations and manual surveys of the orchards. However, these models are highly dependent on the data available; therefore, their prediction capabilities extend only to the volume and quality of their training datasets.

For ‘Hayward’, Morley-Bunker and Salinger [17] found that flowering can be quantified from a thermal time model with budbreak, but such a relationship diminishes with increased temperature. A weighted average of chilling and warming temperatures based on mean daily temperature has been successfully used for predicting budbreak [18]. Snelgar et al. [19] showed that mean temperatures in May, June and July, as well as mean annual temperature, site and latitude, all contributed significantly to predicting the number of flowers per winter bud. It was also found that the number of king flowers per winter bud was strongly associated with low winter temperatures [20]. These studies used historical data, ranging from 1996 to 2013, to train and validate the models for single or multiple cultivars. Since the models use averages for the independent and response variables, the variation and uncertainty analysis, especially in the short or midterm weather events, remains ambiguous [15,18,21,22].

There are strong reasons to revisit modeling flowering timing in kiwifruit, as influenced by regional and weather effects. To date, there are still few reliable models for accurately predicting the date of kiwifruit flowering (but see [23]). This is due to the challenges of systematically recording kiwifruit budbreak and flowering data, as well as obtaining high-resolution weather data at or close to the study orchards. In addition, the kiwifruit industry continually evolves by introducing new cultivars, e.g., *Actinidia chinensis* var. *chinensis* ‘Zesy002’ [21], and emerging management techniques and guidelines. Changes in weather patterns have been more frequent in the last two decades and are expected to become more extreme as a result of climate change in the future [24]. These factors, combined with the increasing demand in the industry for decision-making support tools, motivate us to develop prediction models adaptive to the evolving dynamics of kiwifruit cultivation and weather patterns within the confidence levels that are reasonable to the industry.

Our study sought to develop temperature-based phenological models to predict the date of budbreak and flowering of ‘Hayward’ and ‘Zesy002’ vines in Kerikeri and Te Puke —two important kiwifruit-growing regions in New Zealand—using the historical data collected in those regions from 2007 to 2022 (Figure 1). We employed both classification and regression models to predict the budbreak and flowering dates based on the weather data. In the regression model, we tested the model with linear combinations of temperature, region and cultivar, as well as hierarchical models that use the predicted beginning of budbreak as an input variable for flowering prediction. Several model selection and validation techniques were applied to the series of candidate models. The best performing models were selected in terms of efficiency as a balance in fitting the observational data and with a narrow prediction range, as they are intended to be applied as a useful tool for optimizing management practices.

The rest of this paper is organized as follows: Section 2 describes the description of the training data and the methodology of the model development. The models’ employment, validation, and selection results are given in Section 3, followed by the discussion of practicality and methodology limitations in Section 4. Concluding remarks are made in Section 5.

## 2. Results

### 2.1. Flowering Time of Observational Data

The first flowering dates as day of the year ranged between day 294 and day 348 across both cultivars and regions over the period of the study (Figure 2). The mean day of year for first flowering across all studies was 323.3 ± 1.6 day. Spring is in September in New Zealand.

ANOVA results suggested the mean day of the year of flowering was different between the two cultivars, with ‘Zesy002’ (313.1 ± 1.6 days) flowering significantly earlier than ‘Hayward’ (329.9 ± 1.0 days) in both regions (F_(1, 37)_ = 144.9, *p* < 0.001). However, there was no significant effect of region (F_(1, 37)_ = 2.0, *p* = 0.17) or the region by cultivar interaction (F_(1, 37)_ = 1.3, *p* = 0.26).

The overall mean window for the observation of first flowering was 18.4 ± 2.1 days. To examine the differences in the range of days between the first flowering, a two-way ANOVA was also conducted to assess the effects of region, cultivar, and their interaction on the range variability. The analysis revealed a significant effect of region on the range of first flowering days (F_(1, 37)_ = 5.3, *p* = 0.03), suggesting that the range of first flowering time in Kerikeri (20.4 ± 1.3) was significantly larger than in the Te Puke region (15.3 ± 1.8). In contrast, cultivar alone did not significantly affect the ranges of first flowering time (F_(1, 37)_ = 0.4, *p* = 0.52), nor was there a significant interaction effect between region and cultivar (F_(1, 37)_ = 0.2, *p* = 0.67).

### 2.2. Budbreak Time Prediction

Across both regions and cultivars, a binomial model consistently provided a more efficient and suitable fit for predicting budbreak based on degree day accumulation, with low mean Akaike Information Criterion (AIC) and Bayesian Information Criterion (BIC) values for all categories (Appendix A). The Poisson and polynomial models resulted in much higher values of AIC and BIC, indicating a less optimal fit for the budbreak count data. We selected and used the binomial model for predicting the beginning of budbreak (5%) for each individual vine (Appendix A). Mean AIC values from the binomial model were lower in vines in Te Puke (6.4 and 7.1 for ‘Zesy002′ and ‘Hayward’, respectively) than those of Kerikeri (7.3 and 7.9 for ‘Zesy002′ and ‘Hayward’, respectively), suggesting a more robust model fit to the Te Puke observations. In general, there was a strong relationship between budbreak and GDD (r^2^ > 0.74 for all regions and cultivars; Appendix A), with an r^2^ of 0.88 in ‘Hayward’ vines in Te Puke. Predictions of degree day accumulation at 5% budbreak were made for each vine to incorporate into the final models predicting flowering time.

### 2.3. Flowering Time Prediction

#### 2.3.1. Classification Models

Classification models tended to predict flowering poorly for both ‘Zesy002′ and ‘Hayward’ vines using weather data of degree days and chilling unit accumulation since 1 September (Table 1). A random forests model was the best performing model across the three classification models, and it had the highest accuracy in predicting flowering of ‘Hayward’ vines in the Te Puke region (93.6%), with 6.4% false positive prediction, and no false negative prediction. However, the same random forests model produced a high false negative prediction in the Kerikeri region for both ‘Zesy002’ (23.8%) and ‘Hayward’ (58.5%) vines.

#### 2.3.2. Regression Models

The model with the lowest AIC among the candidate models with full interaction term contains cultivar, degree day accumulation at predicted 5% budbreak and degree day accumulation on 1 September (i.e., the CV.BB.DD model) (Table 2). The models that included full interaction terms between the predictive variables were compared first, and the model with the highest efficiency was subsequently refined using a stepwise process. The final model had a R-squared value accounted for 85.4% of the observed variation in first flowering date in the training data from 2007 to 2019. For the final model, the root mean squared error (RMSE) was 4.1 and mean absolute error (MSE) was 7.1:𝑓(CV,DD,DD) = 312.7 + 15.1 × CV + 10.9 × BB − 3.3 × DD − 3.5 × CV:BB + 1.0 × BB:DD(1)

Here:*f*(CV,DD,DD) denotes the function for calculating the first flowering date.CV is a binary variable for cultivar. CL equals 1 if it is ‘Hayward’.BB is the degree day accumulation at predicted 5% budbreak.DD is the degree day accumulation on 1 September.

The final model has a narrow 99% prediction interval compared with the other candidate models, of between 21.4 and 23 days (Table 2). This is a desired feature that indicates that the final model also has the least uncertainty in the prediction. We compared the predicted date of first flowering on a vine in different regions for ’Zesy002’ and ‘Hayward’ cultivars with the observed testing data (Figure 3, Table 3). The mean first flowering data observed were within the range of prediction. With the exception of ‘Hayward’ at Kerikeri in 2022, most predicted first flowering dates were within a week of the observed mean first flowering from the field. The observed first flowering dates in testing data ranged from 18.7 to 30 days between 2020 and 2022 (see Appendix A), and all the predicted mean first flowering dates fell within the observation window. When the date of budbreak was known, 87.1% of control plants in 2020–2022 began flowering within a narrower 99% prediction interval (19.6–20.8 days) using the cultivar and degree day accumulation as key predictors. When date of budbreak was unknown, 90.6% of plants flowered within a larger 99% prediction interval (30.2–32.5 days) (Appendix A). The k-fold cross-validation resulted in a sample size for each training set of 80% of all data. The overall RMSE was 4.1 (range from 3.3–5.2), with a R-squared of 0.85 (range from 0.75–0.90), and the MSE was 3.1 (range from 2.6–3.5).

### 2.4. Flowering Time Prediction with Position Information

Once budbreak was known, then flowering time could be easily predicted (Appendix A), with or without spatial information. Prior to budbreak, degree day accumulation on 1 September of that year also provided reasonable predictions, especially if the spatial information was known (Table 4). Prediction accuracies were similar when the data were a subset of those that included the position within an orchard.

The model with the lowest AIC among the candidate models with full interaction term contains cultivar, degree day accumulation at predicted 5% budbreak, and degree day accumulation on 1 September (i.e., the CV.BB.DD model) (Table 4). We then subsequently refined the model using a stepwise process, and the final model had a R-squared value accounting for 84.3% of the observed variation in the first flowering date in the training data from data prior to 2019. For the final model, the root mean squared error (RMSE) was 4.0 and mean absolute error (MAE) was 8.3. For this subset of the data where the all the observations were collected from the same orchards, the final model was:𝑓(CV,BB,DD) = 471.2 + 32.1 × CV + 0.09 × BB − 0.61 × DD − 0.06 × CV:BB(2)

Here:*f*(CV,BB,DD) denotes the function for calculating the first flowering date.CV is a binary variable for cultivar. CL equals 1 if it is ‘Hayward’.BB is the degree day accumulation at predicted 5% budbreak.DD is the degree day accumulation on 1 September.

The modeling results indicated that prediction accuracies were higher when using only data that do not include row position within an orchard, but only with cultivar, degree day accumulation and budbreak as the explanatory variables (Figure 4). A share of 98.0% of actual observed first flowering of vines fell into the range of 19.6 to 20.8 days that the model predicted. Interestingly, including row information increased the predicted 99% interval of first flowering. The predicted range of the testing data is shown in Table 5. The k-fold cross-validation resulted in a sample size for each training set of 80% of all data. The overall RMSE was 4.3 (range 3.1–6.0), with a R-squared of 0.87 (range 0.78–0.95), and the MSE was 3.7 (range 2.3–5.1).

## 3. Discussion

Our results suggested that the day of first flower opening can be reasonably predicted using degree day accumulation from an arbitrary date early in the growing season (here 1 September for New Zealand) with a linear model. We found that the best prediction of flowering was achieved using a hierarchical modeling approach with estimated degree day accumulation at the beginning of budbreak based on cultivars. However, if the beginning of budbreak was not known, using an arbitrary date such as 1 September could still give a wider prediction range into which most observations fell. Including the positional information where row and bay were known within the orchard did not improve the model’s predictive power when using longitudinal data from the Kerikeri study orchard. This was probably caused by over-fitting, although including row and bay as predictive variables where available helped to reduce the predicted range of first flowering in all the models we tested.

### 3.1. Cultivar and Regional Difference in Flowering Date

We found that, in general, the mean first flowering date was earlier in ‘Zesy002’ than in ‘Hayward’ vines, which was in line with the results from several previous studies [16,25]. Interestingly, the window/range of first flowering differed significantly between the two study regions but not between the cultivars, and vines in the Te Puke region had a narrower time window for the appearance of the first flower. We interpret this as a result of the higher winter chilling experienced by vines in Te Puke region and hence less variation in budbreak and flowering. These results suggested the weather in the study regions may play a more important role in the variations of flower development, and hence the timing of first flowering [16]. Growing regions with less winter chilling might expect first flowering dates spread through a longer time window, whereas the cultivar did not appear to modulate the range of when the first flowers can be found in the orchards. This also explains the differences in the predictions of the classification model (i.e., the random forest classification) between the two regions (Table 1). In Kerikeri, the first flowering times had larger variances between orchards, and the times were spread across a longer period of time than those the model had predicted, resulting high false negative prediction rates. This result implied that the plant phenological responses to temperature might be more complex and vary between study sites. Weather scenarios in the Kerikeri region, and the metrics such as degree day and chilling unit accumulation, might not be able to fully capture such variation [26].

From the regression model, the period during which 99% of vines in 2020 and 2022 were predicted to start flowering (the 99% prediction interval) was within a range of 21.4–23.0 days, which was similar to the window of first flowering in the observational data (see Figure 2 and Table 4). All candidate regression models included cultivar, and 5% budbreak estimation gave prediction intervals within 25.1 days. The modeling results suggested that there was a consistent relationship between the start of budbreak, cultivar and flowering date, and more complex model that included additional climate information such as accumulated GDD and CU, or daylength, might provide a slightly better fit to the observational data, although that also introduced variation in model predictions. Hall and Snelgar [20] reported that winter temperature had a greater effect on the time of budbreak than temperature in spring. Our findings were consistent with this: for both ‘Hayward’ and ‘Zesy002’, the winter temperature had a strong effect on the beginning of budbreak, and sequentially determined the timing of first flowering. Reasonable predictions can be made at the time of the budbreak without knowing the temperature before the flowering, using the baseline model with only budbreak prediction and cultivar (R^2^ = 0.81) (Table 2).

### 3.2. Model Selection and Validation

The results from the final models highlighted the key predictors in estimating kiwifruit first flowering date: thermal accumulation, especially degree day accumulation, and estimation of budbreak dates. The best model selected remained the same with the entire dataset, as well as with the data collected from a single orchard (Kerikeri study orchard). In the model selection processes, we first evaluated the model performance by accessing interpolation errors which measured how well the model fitted to the data that were used to create the models. Then we compared the model predictions with testing data from the recent years that were not used in the model fitting, to access the extrapolation or forecast errors. The combined approach allowed us to test the model’s ability to generalize to new, unseen data, ensuring that the final model was not only tuned to the historical data, but also robust and effective in making accurate predictions about future trends. The model selection results aligned with research on phenology modeling of kiwifruit and other temperate free trees, where the integration of detailed climatic and site-specific information reflected by the budbreak estimation was used as the main explanatory variable for flowering [24,27,28].

Predicting flowering timing can be difficult because all time-sensitive observational data are inherently correlated, and in this case, it means that plants that flower later naturally experience a higher heat sum just because they were later in the season. We attempted to address this issue by calculating heat sum and chilling sum on a certain day, e.g., 1 September. Thus, we did not have the issue of plants measured late in the season having later heat sums. There are differences between cultivars and regions in the plants’ flowering responses as a result of heat sum, suggesting some degree of causation (Figure 4). It should be noted that heat sum is correlated with many other measured weather data—cold years will have higher chill unit accumulations as well as lower heat sums. One obvious issue with the study was that the same orchards were used across many years, and so testing and training data were not independent. As a result, testing and training data share many of the same vines. Additionally, a great deal of the predictive power came from training the datasets on the same orchard, such as the Kerikeri study orchard. It is likely that new orchards or orchards that were less represented in the data had poorer prediction accuracy. In this situation, other machine learning methods such as transfer learning are likely to have an advantage [29]. This motivated our decision to predict in a scenario most closely related to the question an orchardist would ask: If I know the flowering times of the vines in my orchard, can I use these data to accurately predict when they will flower next year? A more robust model would have independent orchards in the testing and training sets. This would allow for better predictions within a region. Gathering reliable flowering data is expensive and time-consuming, and despite being imperfect, our dataset was still a valuable resource to provide insight into the design of such monitoring activities. We also tested the sensitivity of our model to training/testing data splits by performing k-fold cross-validation. The cross-validation results indicated the regression model was generalizable across different resampling schemes and able to make reliable predictions.

Interestingly, including row and bay information increased the predicted 99% interval of first flowering (Table 4), suggesting that individual vines’ variation or vines at the same position were not consistently early or late flowerers across different years within an orchard. However, with a subset of data from a single orchard, the same-structured regression model can give a higher prediction accuracy with a narrower prediction range even when row and bay information was ignored, compared with the model prediction intervals using full data collected from 19 orchards. This suggested that local conditions within a single orchard, such as microclimate, soil characteristics and specific cultivation practices, played a substantial role in influencing flowering times at an orchard level. Such localized factors may be diluted or confounded when data from multiple orchards are combined. Our model can generate a narrower prediction range with a high accuracy, which are desired features for kiwifruit orchard management [25]. These results highlighted the value of collecting longitudinal data from the same orchard over years and using them in modeling flowering dates. Additionally, integrating precise, onsite weather information through sensors, rather than relying on the nearest weather stations, can further enhance the predictive accuracy. Incorporating more precise weather information into the model can provide insights into impacts from temperature fluctuations over the years on the development of kiwifruit budbreak and flowering.

### 3.3. Challenges and Restrictions

This study relied on data collected by numerous researchers over the last two decades. While the manual sampling and recording methods were consistent, the natural variability in observational data might influence the modeling outcomes. For example, the recorded timing of budbreak lagged behind the start of the biological process, as budbreak needed to be visible before it was identified and recorded in the survey carried out every 3–4 days over the budbreak period. Such discrepancies might result in some earlier occurrences of budbreak not being recorded at its first appearance, potentially skewing the model outcomes, and we applied a logistic regression in budbreak prediction to minimize such an effect.

Currently, kiwifruit management practices includes applying budbreak enhancers such as HC to stimulate uniform flowering [30]. The data used in this study were collected from vines that did not receive any chemical treatment, and hence the first flowering dates were naturally more variable between vines. A critical aspect to consider when evaluating the precision of a phenological model is the variations in the data upon which the model was optimized. In our case, the precision of our predictions was inherently tied to the range and consistency of the observational data available, and the model prediction ranges matched with the range of first flowering observations from the observational data (Table 3 and Table 5).

In more recent years, we observed a shift in the patterns of flowering times. Such a shift seemed to be closely associated with the change in recent weather patterns. The ranges of first flowering date were broader in the last three years than in historical records, and the mean flowering dates were earlier, co-occurring with the warmer winters from 2020 to 2022. These shifts could signal that the historical data and assumptions used to calibrate our models—such as the fixed critical temperature of 7.0 °C and a 1 September cut-off for tracking phenological changes—might no longer provide an accurate framework for prediction. Future work should focus on revisiting and possibly revising these foundational model assumptions and parameters, such as exploring alternative critical temperature thresholds and cut-off dates, and/or incorporating a variance structure in the model to better account for increased variability and the impacts of changing climatic conditions on flowering patterns.

## 4. Materials and Methods

### 4.1. Data

Data collection for this study included two major tasks: firstly, reviewing historical studies and collecting data on the budbreak and flowering development and timing; secondly, collecting related historical weather data, ambient temperature in particular, for the sites where the budbreak and flowering data were collected.

#### 4.1.1. Budbreak and Flowering

Detailed data on the development of budbreak and flowering of both ‘Hayward’ and ‘Zesy002’ kiwifruit vines were collected from vines used in previous experiments from 2007 to 2022. The data were collected from untreated vines in a range of different studies examining the effects of a numerous management treatments (e.g., budbreak enhancers, girdling, thinning) on vine phenology and subsequent production. These data were collected across 19 study orchards in both Kerikeri and the Bay of Plenty/Te Puke. The data collected depended on the key question of each study.

In these experiments, the time of budbreak and time of flowering were consistently monitored on selected canes at 3- to 4-day intervals. A bud was termed as broken when it was approximately 10 mm in length with a small amount of green tissue visible (defined as advanced budburst by Brundell [31]). A flower was determined to be open when there was sufficient access for bees. Some studies recorded vine-level metadata such as row and bay position within the orchard. When these were available, we used this subset of data to investigate within-orchard effects. We performed a two-way ANOVA to assess the effects of region, cultivar and their interaction on the mean day of the year that flowers were first observed to be open.

#### 4.1.2. Weather Data

The vine data on budbreak and flowering were combined with climate data from the weather station closest to each experimental site. Data from the weather station closest to each study location were extracted from the MetWatch (http://www.metwatch.co.nz (accessed on 20 March 2024)) and NIWA (www.cliflo.niwa.co.nz, via R package “clifro” (accessed on 4 April 2024)) weather databases, and when multiple stations were available, the station with the most complete data was used. Figure 1 shows the locations of the two main study regions.

To acquire weather data, we searched open stations within a 30 km radius of each orchard from the NIWA climate database. Hourly temperature data from these stations were then extracted and used to calculate the two main weather metrics: (1) growing degree day accumulation, by taking the average temperature for a day, subtracting a base temperature of 7 °C, and then summing up the daily values from 1 July until 30 June of the next year; and (2) chilling unit accumulation, calculated by counting the number of hours at or below a temperature of 7 °C from 1 January to 30 December of the same year. In cases where there were no nearby open stations or there were open stations without hourly temperature data, we obtained GDD and CU accumulation information from the MetWatch weather and disease platform, using the two HortPlus weather stations at Kerikeri and the Te Puke, respectively. In addition, based on the latitude of each site, daylength on 1 September each year was also extracted for use as a potential parameter in the model.

### 4.2. Methodology

#### 4.2.1. Predicting 5% Budbreak

Budbreak and flowering data were surveyed at regular intervals; thus, the first recording of budbreak or flowering may not have been the actual first event. To use budbreak as a predictor for flowering, we fitted regression models to the budbreak counts to predict the degree day accumulation at 5% budbreak. We applied three types of regression models for the relationship between degree day accumulation and budbreak of the two cultivars in the two study regions. We fitted (1) a binomial model fitting the proportion of total budbreak; (2) a Poisson model; and (3) a polynomial model for the count event of the budbreak against degree day accumulation to each vine per annum. We then calculated the degree day value corresponding to 5% of the total budbreak given by the best fitting model. Model performance was assessed by Akaike Information Criterion (AIC) values of the three models. Degree day accumulation at 5% budbreak was then incorporated as a predictor into the hierarchical models to predict flowering time.

#### 4.2.2. Predicting Day of First Flowering

The aim was to model flowering time that had both high prediction ability and high explainability. To examine this, we split the data into test and training sets. However, much of the data were collected on the same orchards over many years, so the assumption of independence between the two sets of data was violated. Because we were most interested in predicting the flowering time of the current year’s vines, we wanted the test set to mirror the key question most accurately. In this way, we trained models on all data from 2007 to 2019 and then tested these models with 2020 to 2022 data. Assessment accuracy was assessed as the proportion of false positives and negatives, as well as the fit of the observed values versus those predicted.

A range of statistical learning modeling techniques were used to examine the relationship between time of flowering and weather or location predictors. The response of flowering was treated two different ways:Presence or absence of flowering (i.e., classification models).The day of year when the first flower was observed (i.e., regression models).

Models predict flowering very well if the cumulative heat sum of that day is known, owing to the inherent confounding effects of time on of both degree day accumulation and flowering. To try to address this problem, we used predictors that were based on a particular day, rather than accumulation over time. We tried subsets of predictors, including:The GDD accumulated at 5% budbreak at the nearest weather station.Degree day accumulation from the nearest weather station on 1 September of that year.Chilling units below 7 °C accumulations [32] from the nearest weather station on 1 September of that year.Daylength on 1 September of that year.Predicted degree day accumulation for 5% budbreak (i.e., once budbreak is established, does it predict flowering time?) [14].

In addition, cultivar (‘Hayward’ or ‘Zesy002’), region (Te Puke or Kerikeri), and orchard were used as predictors in the models.

Models for classification investigated included random forests using the R packages randomForest [33] and ranger [34], and logistic regression on the presence/absence of flowering. These models were chosen for their respective strengths: binomial glm for its simplicity and interpretability in binary classification, and the ranger and randomForest packages for their robust, ensemble-based approach to handling complex data. Linear regression was used to predict the first day of flowering. Orchard was not used as a predictor in the classification models because of the incomplete overlap between training and testing datasets (see Appendix A). All numerical predictors were scaled and centered for classification models.

#### 4.2.3. Predicting Day of First Flowering Where Locational Information Is Known

To examine whether spatial information could help to predict the within-orchard variation in the first flowering day, we analyzed the subset of orchards where row information was known. We focused on orchards that were repeatedly sampled over many years, as row information will reveal microsite differences. We therefore limited the data to one study orchard at Kerikeri (Orchard 4 as in Figure 2 and Appendix A), where repeated measurements were carried out in 2013–2015 and 2018–2020. The models were rerun on this subset of data, including the spatial location within an orchard block, i.e., row and bay position. The model predictions were then compared with the test data, and we also compared the predictive accuracy of the models to the version with the same model but omitting the row and orchard interaction. The final model was cross-validated with the training data, and the prediction accuracy was assessed by comparison with the testing data.

#### 4.2.4. Model Selection

Model selection of classification models was based on lower AIC values for evaluating model fit and complexity. The model selection of regression models was based on the model with high efficiency, i.e., low AIC and coefficient of determination (R^2^) value on the training data from 2007 to 2019. The models that included full interaction terms between the predictive variables were compared first, and the model with highest efficiency was subsequently refined using a stepwise process. The final model was then used for predicting the first flowering date in 2020–2022, and the predictions were compared with the observational data (i.e., the testing data). The percentage of observed flowering within the 99% prediction interval range was calculated as the intervals by adding and subtracting the critical value of the standard normal distribution corresponding to the 99% confidence level (2.576) times the standard errors from the predicted values. K-fold cross-validation was applied to the final model to assess the performance of the final model and its sensitivity to the split of training and testing data [35]. Given our model was developed based on data from earlier years and then predicted for the more recent year, the cross-validation could flag issues such as overfitting or selection bias, and it provided insights into how the model generalized to testing data that were not bounded by a specific range of study year.

All analyses were conducted in R (R Core Team 2021 [36]). Unless otherwise stated, all statistical values are reported as mean ± standard error (*SE*).

## 5. Conclusions

In this study, we developed a hierarchical model for predicting first flowering of two kiwifruit cultivars by combining knowledge of kiwifruit phenology, meteorology and data science. This interdisciplinary approach enabled us to refine our understanding of plant behavior in response to weather conditions by using a longitudinal dataset of budbreak and flowering over the last two decades to develop models to predict kiwifruit flowering times. We tested candidate models that included information on cultivar, CU, GDD, day length, and site location. The final model identified had a hierarchical structure, where the beginning of budbreak date was first predicted and then used sequentially for prediction of the first flowering day. The model demonstrated high predictive accuracy and can be generalized for prediction for future seasons. This capability will allow enhanced planning and operational managements in kiwifruit cultivation.

## Figures and Tables

**Figure 1 plants-13-02231-f001:**
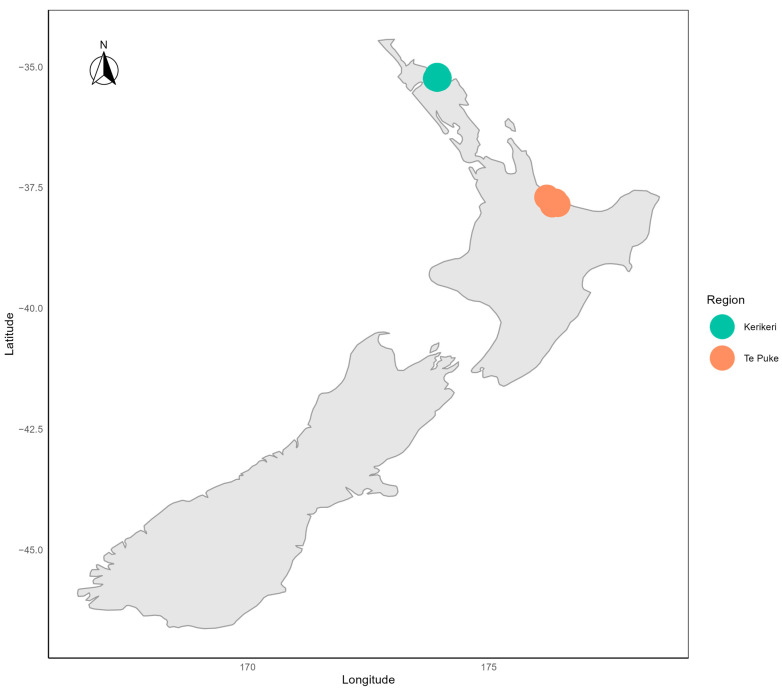
The location the two New Zealand study regions. The points on the plot show the rough locations of the kiwifruit orchards where data were collected.

**Figure 2 plants-13-02231-f002:**
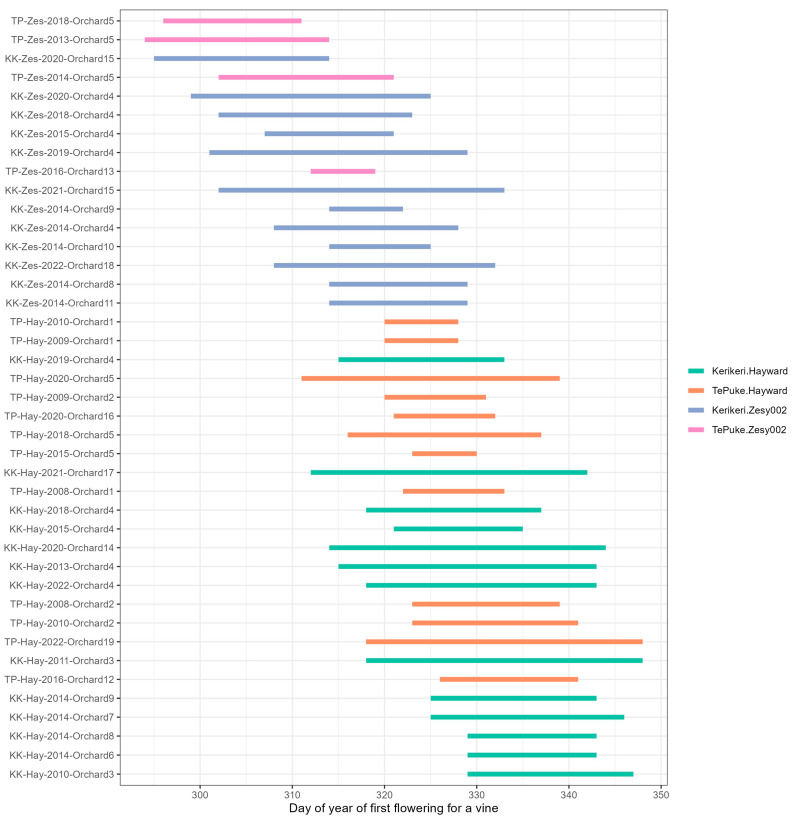
The range of first flowering days (as day of the year) for vines of both kiwifruit cultivars in different orchards/regions over years. Spring is in September in New Zealand. From the top to the bottom, the data were ordered based on the mean day of year of the first flowering for the vine. The first two letters on the orchard name indicate the region of the orchards (KK = Kerikeri, TP = Te Puke), followed by the cultivar (Hay = ‘Hayward’, Zes = ’Zesy002’), study year, and the orchard names, which are anonymized and labeled by numbers.

**Figure 3 plants-13-02231-f003:**
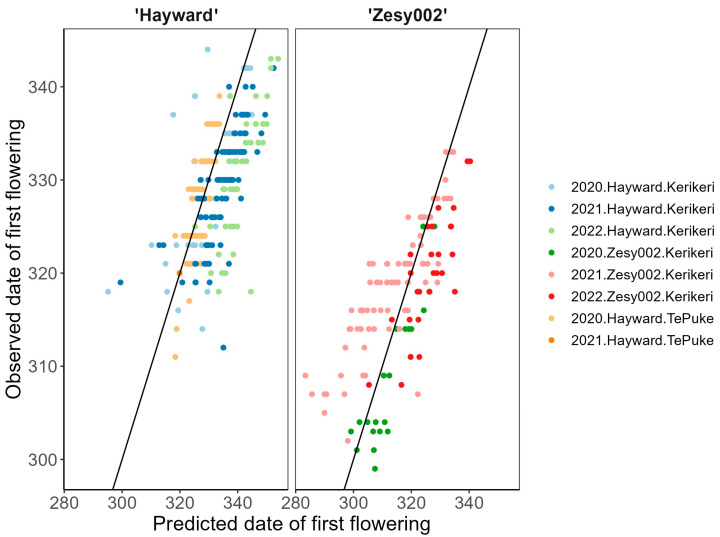
Observed day of first flowering for a kiwifruit vine versus predicted by the various combinations of predictive variables from the best model. The predictive variables included in the model are cultivar, predicted 5% budbreak date, and degree day accumulation on 1 September. The solid line depicts a 1:1 relationship.

**Figure 4 plants-13-02231-f004:**
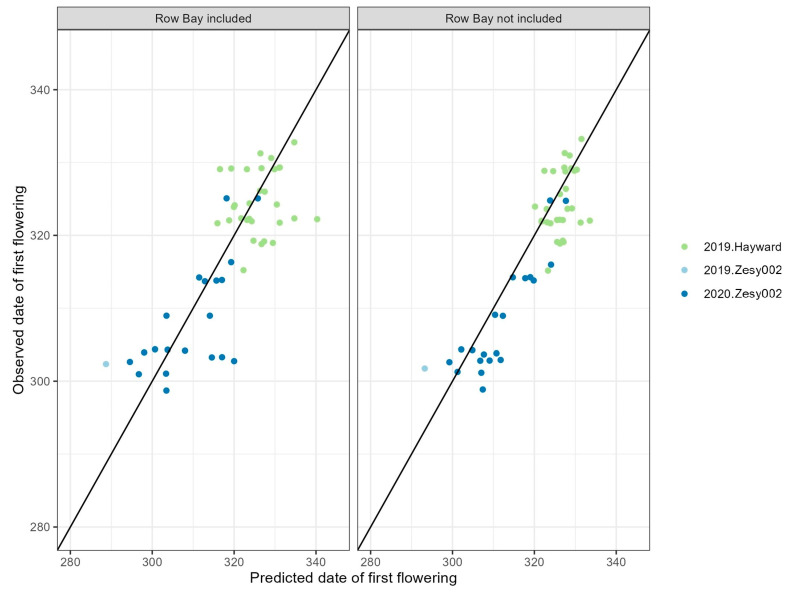
Observed day of first flowering for a kiwifruit vine versus predicted for control plants in orchards where row and bay position was known in the Kerikeri study orchard (Orchard 4 as in Figure 2 and Appendix A). The solid line depicts a 1:1 relationship.

**Table 1 plants-13-02231-t001:** Percentage prediction accuracy, sensitivity (true positives, where prediction = 1 and flowering = 1) and specificity (true negatives, where prediction = 0 and flowering = 0) of classification of kiwifruit flowering based on weather data. Flowering is the presence or absence of flowering on a date, prediction is the predicted flowering on that date. Models were trained on data from 2007 to 2019 and predictions made for 2020 and 2022. The predictions were compared with observations from 2020 and 2022. The table also shows false positives (prediction = 1 but flowering = 0) and false negatives (prediction = 0 but flowering = 1). False negatives and false positives are shaded in grey. Binomial glm = classification from binomial glm, ranger = classification using random forests using the ranger package in R, random forests = classification by random forests using the random forest package in R.

Region	Cultivar	Prediction	Flowering	% Classification		
				Binomial Glm	Ranger	Random Forests
Kerikeri	‘Zesy002’	0	0	0	0	0
Kerikeri	‘Zesy002’	0	1	33.0	23.8	23.8
Kerikeri	‘Zesy002’	1	0	0	0	0
Kerikeri	‘Zesy002’	1	1	67.0	76.2	76.2
Kerikeri	‘Hayward’	0	0	0	0	0
Kerikeri	‘Hayward’	0	1	65.8	60.9	58.5
Kerikeri	‘Hayward’	1	0	0	0	0
Kerikeri	‘Hayward’	1	1	34.2	39.1	41.5
Te Puke	‘Hayward’	0	0	4.1	0	0
Te Puke	‘Hayward’	0	1	29.6	0	0
Te Puke	‘Hayward’	1	0	2.3	6.4	6.4
Te Puke	‘Hayward’	1	1	64.0	93.6	93.6

Note: The total number of predictions/observations for each region and cultivar was: Kerikeri ‘Zesy002’ *n* = 743; Kerikeri ‘Hayward’ *n* = 641; Te Puke ‘Hayward’ *n* = 1045.

**Table 2 plants-13-02231-t002:** The range between minimum (Min) and maximum (Max) 99% prediction intervals (in days) for the 2020–2022 kiwifruit plants, and the percentage of plants that actually flowered within those intervals. CV = cultivar, BB = degree day accumulation at predicted 5% budbreak, DD is degree day accumulation on 1 September, CU is chilling unit accumulation on 1 September, DL is daylength. The CV.BB.DD_1 (bolded) was selected as the best model. The selected best model and the modeling outputs are in bold font.

Model	99% Prediction Interval (Days)		Percentage of Observed Flowering within the Range of Prediction (%)	AIC	R^2^
	Min	Max			
CV.DL	31.7	31.7	95.9	7905.4	0.68
CV.DD	35.3	36.2	94.5	8170.2	0.60
CV.BB	24.2	24.7	92.6	7248.7	0.81
CV.CU	35.4	35.5	97.9	8181.3	0.60
CV.BB.DD	21.4	31.7	85.7	6953.7	0.85
**CV.BB.DD_1**	**21.4**	**23**	**87.1**	**6950.5**	**0.85**
CV.BB.CU	23.7	25.1	92.6	7206.1	0.82
CV.BB.DL	23.1	24.4	87.8	7137.2	0.83
CV.DD.DL	30.2	32.5	90.6	7796.8	0.71
CV.CU.DL	30.9	31.1	96.6	7846.9	0.70

**Table 3 plants-13-02231-t003:** The mean flowering day predicted by the selected model for 2020 to 2022 compared with the observations as the test data. The predictive variables included in the model are kiwifruit cultivar, predicted 5% budbreak date, and growing degree day accumulation since 1 September.

Year	Region	Cultivar	Flowering Day of the Year			
			Predicted	Observed	Difference	Range of Prediction
2020	Kerikeri	’Zesy002’	312.0	306.4	5.6	301.2–326.9
2020	Kerikeri	‘Hayward’	330.4	330.2	0.2	319.6–343.6
2020	Te Puke	‘Hayward’	326.6	326.7	−0.1	315.9–339.8
2021	Kerikeri	’Zesy002’	314.0	319.0	−5.0	303.2–328.9
2021	Kerikeri	‘Hayward’	334.5	329.1	5.5	323.8–347.8
2022	Kerikeri	’Zesy002’	325.8	319.3	6.5	309.2–337.8
2022	Kerikeri	‘Hayward’	340.8	330.5	10.3	314.9–354.3
2022	Te Puke	‘Hayward’	328.4	332.2	−3.7	312.1–344.8

**Table 4 plants-13-02231-t004:** The range of minimum (Min) and maximum (Max) 99% prediction intervals (in days) for the 2020–2022 control kiwifruit plants, and the percentage of plants that actually flowered within those intervals. CV = cultivar, BB = predicted 5% budbreak, DD is degree day accumulation on 1 September, CU is chilling unit accumulation on 1 September, DL is daylength. Models are compared both including and excluding (gray-shaded) row and bay position information. The selected best model and the modeling outputs are in bold font.

Model	Row Bay	99% Prediction Interval (Days)		% Observed Flowering within the Range of Prediction	AIC	R^2^
		Min	Max			
CV.BB	included	19.7	27.7	80.4	302.1	0.84
CV.BB	excluded	20.0	21.1	92.2	295.7	0.83
CV.BB.CU	included	18.7	28.3	74.5	302.1	0.84
CV.BB.CU	excluded	19.3	22.9	84.3	292.3	0.84
CV.BB.DD	included	20.5	52.6	96.1	302.1	0.84
CV.BB.DD	excluded	19.8	22.4	98.0	292.3	0.84
CV.BB.DD_1	included	20.1	32.0	96.1	302.1	0.84
**CV.BB.DD_1**	**excluded**	**19.6**	**20.8**	**98.0**	**292.3**	**0.84**
CV.CU	included	36.8	49.7	100.0	322.7	0.76
CV.CU	excluded	33.3	33.7	98.0	331.5	0.65
CV.CU.DL	included	37.7	62.7	100.0	322.7	0.76
CV.CU.DL	excluded	31.2	40.7	62.7	331.5	0.65
CV.DD	included	36.9	75.4	98.0	322.7	0.76
CV.DD	excluded	32.8	33.3	94.1	331.5	0.65
CV.DD.DL	included	44.7	141.4	100.0	322.7	0.76
CV.DD.DL	excluded	31.2	37.7	41.2	331.5	0.65
CV.DL	included	37.4	52.8	100.0	322.7	0.76
CV.DL	excluded	33.0	33.5	100.0	330.7	0.65

**Table 5 plants-13-02231-t005:** The mean flowering day predicted by the selected model for the single kiwifruit orchard at Kerikeri (Orchard 4 as in Figure 2 and Appendix A) comparing the observations within the test data. The predictive variables included in the model are cultivar, growing degree day accumulation at predicted 5% budbreak date, and degree day accumulation on 1 September.

Year	Cultivar	Flowering Day of the Year			
		Predicted	Observed	Difference	Range of Prediction
2019	‘Zesy002’	305.1	317.2	−12.1	291.4–318.8
2019	‘Hayward’	330.2	324.5	5.7	316.4–344.1
2020	‘Zesy002’	316.6	308.5	8.2	302.6–330.6

## Data Availability

The datasets presented in this article are not readily available because the data were collected as part of commercially funded research. Requests to access the datasets should be directed to the corresponding author.

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
