# Peer review of "A Hierarchical Model to Predict Time of Flowering of Kiwifruit Using Weather Data and Budbreak Dynamics"

_plants, 2024, doi:10.3390/plants13162231_

Round 1

Reviewer 1 Report

Comments and Suggestions for Authors

The title of this paper is “A hierarchical model to predict time of flowering of kiwifruit using weather data and budbreak dynamics”. They developed a predictive model for the flowering time using weather data and observations of budbreak dynamics for the kiwifruit varieties of ‘Hayward’ and ‘Zesy002’. The authors analyzed budbreak and flowering timing alongside GDDs and CU accumulation, and other environmental variables. Next, the authors trained the model with data from 2007 to 2019, and then evaluated the model's efficacy using the testing data from 2020 to 2022. Different regression models identified a hierarchical structure with the accumulation of GDDs at the start of budbreak, one of the key predictors of flowering time. There are some comments provided as following:

1.       The statistical backgrounds should be described clearer. For example, in the section of 2.2. of budbreak time prediction, before the binomial model was used, you check the model assumptions or not?  Also, in table 1, the differences between these classification models should be mentioned, and the setup of the model parameters should be provided. We only knew that different R packages were used in the study.  In tables 2 and 4, how did you achieve the 99% predicton interval?    What is the formula?  Lines 528-529, why the random forests need scaled and centered the variables? Lines 547-548, the statement is wrong.

2.       Lines 539-540, can you provide a diagram to explain the effect of row and bay position in this study?

Author Response

Comment 1: The statistical backgrounds should be described clearer. For example, in the section of 2.2. of budbreak time prediction,before the binomial model was used, you check the model assumptions or not?  Also, in table 1, the differences between these classification models should be mentioned, and the setup of the model parameters should be provided. We only knew that different R packages were used in the study.  

Response 1: The authors thank the reviewer for the feedback. We have now provided some more details on the description and differences between the classification models (Lines 526-528). Default parameters were used from the functions of the R packages in our study.

Comment 2: In tables 2 and 4, how did you achieve the 99% prediction interval?    What is the formula? 

Response 2: To calculate the 99% prediction intervals from a glm regression model, we first fit the model and used the predict function with se.fit = TRUE to obtain the predictions and their standard errors. The 99% prediction intervals are then computed by adding and subtracting the product of the critical value from the normal distribution and the standard errors from the predicted values. This provides the lower and upper bounds of the prediction interval for each prediction. We added the calculation steps of the 99% prediction interval to Lines 559-561 in the methods of the revised manuscript.

Comment 3: Lines 528-529, why the random forests need scaled and centered the variables?

Response 3: Yes, the scale and centre of the variable is indeed not required for random forests. We did the scale and centre according to best practices for consistent preprocessing as a part of our standard workflow for all classification models to facilitate comparison. We have modified the manuscript and removed the contents referring this is specifically done for random forests at Line 532.

Comment 4: Lines 547-548, the statement is wrong.

Response 4: Thanks for pointing this out, we need to specify the different the model selection methods for classification and regression models.

We have revised our explanation to accurately reflect our model selection criteria, emphasizing the use of AIC for evaluating model fit and complexity in classification models, and both AIC and adjusted R² for regression models where applicable at Lines 551-552.

Comment 5: Lines 539-540, can you provide a diagram to explain the effect of row and bay position in this study?

Response 5: Thanks for the suggestion. With Table 4 and Figure 4, we think there were sufficient comparisons between the models with or without row and bay information. Row and bay position the spatial location of where the data was collected from the orchard. The row and bay were added to the model to test whether this is a covariate in the flowering model and can improve the model fit to the training data. However, the larger 99% prediction interval compared to the model where the row and bay information were excluded suggest that the new variable may be introducing additional variability or noise into the model.

Reviewer 2 Report

Comments and Suggestions for Authors

Since the results/conclusions are not new (see also 325-329), it should be better to shorten the all paper.

Minor notes in the attached file

Author Response

Comment 1: Lines 64-65 Unnecessary; Lines 65-66 repletion, see lines 31-32

Response 1: Thanks for pointing out the repetition, we have removed the two sentences.

Comment 2: Lines 570-572 This is not reported in the abstract.Since this results/conclusions are not new (see also 325-329), it should be better to shorten the all paper.

Response 2: The authors respectfully disagree with this comment. In Lines 325-329, we stated in the original manuscript that “Hall and Snelgar [20] reported that winter temperature had a greater effect on the time of budbreak than temperature in spring, and our findings were consistent with this: for both ‘Hayward’ and ’Zesy002’, the winter temperature had a strong effect on the beginning of budbreak, and sequentially determined the timing of first flowering.” – Our finding agrees with Hall and Snelgar (2014) and other previous studies that “winter temperature had a greater effect on the time of budbreak than temperature in spring”. However, the novelty of this study was not only finding effects of temperature on budbreak, but more importantly the sequential determination of flowering time of the studied cultivars from the temperature-driven phenological event. In our manuscript, we have quantified the influence of temperature on flowering by developing a robust hierarchical prediction model, using the heat accumulation at budbreak as a key covariate. These have been reported in the abstract.